# Analysing the Combined Effects of Radiotherapy and Chemokine Receptor 5 Antagonism: Complementary Approaches to Promote T Cell Function and Migration in Oesophageal Adenocarcinoma

**DOI:** 10.3390/biomedicines12040819

**Published:** 2024-04-08

**Authors:** Maria Davern, Cillian O’ Donovan, Noel E. Donlon, Eimear Mylod, Caoimhe Gaughan, Anshul Bhardwaj, Andrew D. Sheppard, Dara Bracken-Clarke, Christine Butler, Narayanasamy Ravi, Claire L. Donohoe, John V. Reynolds, Joanne Lysaght, Melissa J. Conroy

**Affiliations:** 1Cancer Immunology and Immunotherapy Group, Department of Surgery, School of Medicine, Trinity St. James’s Cancer Institute, Trinity Translational Medicine Institute, St. James’s Hospital, Trinity College Dublin, D08W9RT Dublin, Ireland; maria_davern@dfci.harvard.edu (M.D.); ciodonov@tcd.ie (C.O.D.); donlonn@tcd.ie (N.E.D.); mylode@tcd.ie (E.M.); cagaugha@tcd.ie (C.G.); davernma@tcd.ie (A.B.); sheppaa@tcd.ie (A.D.S.); dara.bracken-clarke@nih.gov (D.B.-C.); chbutler@tcd.ie (C.B.); nravi@stjames.ie (N.R.); donohoe.claire@gmail.com (C.L.D.); reynoldsjv@stjames.ie (J.V.R.); jlysaght@tcd.ie (J.L.); 2Dana-Farber Cancer Institute, Harvard Medical School, Boston, MA 02215, USA; 3Cancer Immunology Research Group, Department of Anatomy, School of Medicine, Trinity Biomedical Sciences Institute and Trinity St. James’s Cancer Institute, Trinity College Dublin, D08W9RT Dublin, Ireland

**Keywords:** T cell recruitment, Maraviroc, cancer immune suppression, chemokines, CCR5, oesophageal adenocarcinoma

## Abstract

The presence of an immunosuppressive tumour microenvironment in oesophageal adenocarcinoma (OAC) is a major contributor to poor responses. Novel treatment strategies are required to supplement current regimens and improve patient survival. This study examined the immunomodulatory effects that radiation therapy and chemokine receptor antagonism impose on T cell phenotypes in OAC with a primary goal of identifying potential therapeutic targets to combine with radiation to improve anti-tumour responses. Compared with healthy controls, anti-tumour T cell function was impaired in OAC patients, demonstrated by lower IFN-γ production by CD4^+^ T helper cells and lower CD8^+^ T cell cytotoxic potential. Such diminished T cell effector functions were enhanced following treatment with clinically relevant doses of irradiation. Interestingly, CCR5^+^ T cells were significantly more abundant in OAC patient blood compared with healthy controls, and CCR5 surface expression by T cells was further enhanced by clinically relevant doses of irradiation. Moreover, irradiation enhanced T cell migration towards OAC patient-derived tumour-conditioned media (TCM). In vitro treatment with the CCR5 antagonist Maraviroc enhanced IFN-γ production by CD4^+^ T cells and increased the migration of irradiated CD8^+^ T cells towards irradiated TCM, suggesting its synergistic therapeutic potential in combination with irradiation. Overall, this study highlights the immunostimulatory properties of radiation in promoting anti-tumour T cell responses in OAC and increasing T cell migration towards chemotactic cues in the tumour. Importantly, the CCR5 antagonist Maraviroc holds promise to be repurposed in combination with radiotherapy to promote anti-tumour T cell responses in OAC.

## 1. Introduction

Oesophageal adenocarcinoma (OAC) is the predominant subtype of oesophageal cancer in the Western world [1]. The current standard of care includes surgical removal of the tumour, preceded by neoadjuvant chemoradiation or perioperative chemotherapy [2,3]. Response rates to current treatment strategies remain low at 20–30% [4]. Additional therapies have been approved in the second- and third-line setting, which include targeted therapies such as VEGFR- or HER2-targeting antibodies [5,6]. More recently, immunotherapies such as immune checkpoint inhibitors have been approved in the neoadjuvant setting in combination with chemotherapy and in the adjuvant setting for use as a monotherapy in OAC patients [7,8]. It is well-documented that a higher tumour mutational burden (TMB) correlates with a favourable response to immune checkpoint blockade, and OAC was ranked 5th out of 30 tumour types for having a high TMB [9]. Despite OAC having a relatively high TMB compared with other cancers, response to immunotherapies is typically limited to a subset of patients [10]. A meta-analysis conducted by Chen et al., examining the effectiveness of PD-1, PD-L1, and CTLA-4 immune checkpoint blockade in patients with oesophagogastric junctional adenocarcinoma and advanced gastric cancer, revealed that the addition of immune checkpoint inhibitors to second- and third-line regimens achieved an objective response rate in only 9.9%, 12.0% and 2.1% of patients, respectively [11]. Accumulating evidence pinpointed the tumour microenvironment (TME) as central in mediating poor responses to current chemoradiation and immunotherapy regimens in OAC [12]. This immunosuppressive TME may counteract some of the immunostimulatory benefits conferred by the presence of a high TMB. Many features of the TME shape an immunosuppressive milieu through the direct or indirect inhibition of anti-tumour immune cell function and skewing immune cell recruitment to favour a more abundant pro-tumour immunomodulatory phenotype [12,13].

The chemokine network is comprised of chemokines and their cognate receptors and plays a vital role in shaping immune responses through their classical chemotactic role and their broader biological effects [14]. In the context of the solid TME, chemokines influence the immune contexture through their chemotactic and functional effects on immune cells [15,16,17]. Dysregulated chemokine signalling in the TME supports the malignant outgrowth of tumours, exclusion of anti-tumour immune cells, and abundance of immunosuppressive cells [16,18]. Our previous studies identified impaired migratory capacity of circulating T cells in OAC patients and revealed that OAC tumours had a low infiltration of T cells despite an abundance of Th1 chemokines [19].

Kavanagh et al. previously reported that OAC tumour tissue secreted abundant levels of Th1 chemokines (RANTES and MIP-1α) [19]. However, this did not correspond with the enrichment of tumour-infiltrating T cells expressing these corresponding receptors [19]. Furthermore, circulating T cells from OAC patients exhibited an impaired migratory capacity with decreased levels of Th1-associated CXCR3^+^ cells [19]. Collectively, these studies revealed that T cell infiltration to OAC tumours was compromised and that therapies targeting T cell trafficking could benefit OAC patients. Additional studies also identified that irradiation stimulates tumour cells and immune cells to secrete RANTES and MIP-1α [20], which suggests that irradiation might promote the trafficking of T cells to OAC tumours. In a previous study, we observed that irradiating OAC tumour biopsy explants ex vivo had favourable effects on the secretome, significantly increasing the secretion of anti-tumour cytokines IL-21 and IL-31 [21] and decreasing the production of a tumour-promoting cytokine IL-23 [21]. In addition, radiation induced an anti-angiogenic tumour milieu by reducing the secretion of VEGF-A, BFGF, Flt-1, and PIGF pro-angiogenic factors [21].

In this study, we focused our investigation on the CCR1, CX_3_CR1, and CCR5 chemokine pathways, which have important roles in anti-tumour T cell biology. Radiation forms a principal component of the therapeutic backbone for OAC. Within the irradiation field, T cells that reside in tumour-draining lymph nodes that infiltrate the tumour, and those that are in peripheral circulation can be exposed to radiation, which likely impacts their function [22]. Previous studies highlighted the toxic effects of radiation treatment on lymphocytes, including T cells, as well as the immunostimulatory benefits, such as the enhancement of antigen presentation and increased T cell infiltration to the tumour in many cancer types [22]. This study specifically investigated the immunomodulatory effects of radiation therapy and chemokine receptor antagonism on T cell function and migration in OAC. A key objective was to identify potential therapeutic targets to combine with radiotherapy to enhance anti-tumour responses in OAC.

CCR1 and CCR5 are the known receptors for MIP-1α and RANTES ligands [23]. CX_3_CR1 is the known receptor for fractalkine ligand [24,25]. These chemokine networks play a role in anti-tumour immune cell migration, namely T cell and Natural Killer (NK) cells. CCR1 is expressed on the surface of pro-inflammatory T cells [26]. CCR5 was identified on the surface of regulatory T cells (Tregs) as well as effector T cells [23,27,28,29]. The expression of CX_3_CR1 on CD8^+^ T cells marks antigen-experienced T cells with distinct roles in immune surveillance and homeostasis and correlates with the degree of effector T cell differentiation [24,25].

Our findings support our previous data highlighting dysregulated immune responses in OAC patients [12]. Here, our data reveal that effector T cell function is diminished in OAC patients but could be rescued with clinically relevant doses of irradiation ex vivo. Additionally, irradiated T cells acquired an increased migratory capacity towards the chemotactic cues of the OAC TME. These findings have implications for circulating T cells and lymph node-residing T cells that lie within or close to the radiation field of OAC patients. T cells derived from OAC patients had significantly higher levels of CCR5 compared with healthy controls. Antagonizing CCR5 signalling, with the FDA-approved Maraviroc, in irradiated T cells increased their migration towards the chemotactic cues of the irradiated OAC TME. OAC patient-derived T cells exhibited diminished effector function, and in vitro treatment with Maraviroc enhanced their IFN-γ production, thus further supporting the anti-tumour potential of this antagonist.

## 2. Methods

### 2.1. Ethical Approval

Ethical approval was granted from the St. James’s Hospital/AMNCH Ethical Review Board. All samples were collected with prior informed written consent for sample and data acquisition from patients attending St. James’s Hospital or from healthy age-matched human participants (recruited from employees working in St. James’s Hospital). The World Medical Association’s Declaration of Helsinki guidelines on medical research involving human subjects were strictly followed during this study. In line with GDPR and data protection policies, patient samples were pseudonymized.

### 2.2. Specimen Collection

From 2018 to 2021, treatment-naïve OAC patients undergoing endoscopy at St. James’s Hospital at their time of diagnosis were recruited for this study. A total of 9 OAC patients provided treatment-naïve whole blood samples (7 males and 2 females with an age range of 51–75 and average age of 63.2 years). Moreover, 6 OAC patients provided treatment-naïve tumour tissue biopsies (5 males and 1 female with an age range of 48–75 and average age of 61.0 years). Also, 6 healthy age-matched participants (5 males and 1 female) were also included in this study, with an age range of 55–61 years and average age of 57.8 years. The demographics for all participants are detailed in Table 1.

### 2.3. Generation of Tumour-Conditioned Media

OAC patient-derived tumour-conditioned media (TCM) was generated as previously described [30]. OAC tumour explants of ~2–3 mm^3^ were transferred into 1 mL of serum-free M199 media (Gibco, Billings, MT, USA), supplemented with gentamicin in a 12-well plate. One piece was irradiated with 1.8 Gy, and the other piece was non-irradiated (NIR). The explants were cultured for 24 h at 37 °C, 5% CO_2_. The resulting tumour-conditioned media (TCM) and irradiated TCM (IR-TCM) were harvested and stored at −80 °C until required for further experimentation. All irradiations were performed using an X-Strahl cabinet X-ray irradiator (RS225) (X-Strahl LTD, Walsall, UK).

### 2.4. T Cell Activation

Treatment-naïve OAC donor peripheral blood mononuclear cells (PBMCs) were isolated from whole blood using Ficoll-Pacque (GE healthcare, Chicago, IL, USA) density gradient centrifugation and expanded in RPMI 1640 medium (Gibco, Waltham, MA, USA) supplemented with 10% foetal bovine serum and penicillin–streptomycin (Gibco, Waltham, MA, USA) using plate-bound anti-CD3 (10 μg/mL, Biolegend, San Diego, CA, USA), anti-CD28 (10 μg/mL, Ancell, Bayport, MN, USA), and recombinant human IL-2 (10 IU/mL, Immunotools, Friesoythe, Germany) for 3 days. PBMCs were irradiated with a 1.8 Gy dose of irradiation on days 1 and 2, 24 h apart, or cells were non-irradiated (NIR). All irradiations were performed using an X-Strahl cabinet X-ray irradiator (RS225) (X-Strahl Ltd., Walsall, UK). For experiments that included treatment with antagonists, in the last 24 h of the T cell activation process, PBMCs were treated with 1 nM CCR1 antagonist J113863 (Axon MedChem, Reston, VA, USA), 80 μM of CCR5 antagonist Maraviroc (Axon MedChem, Reston, VA, USA), 245 nM CX_3_CR1 antagonist AZD8798 (Axon MedChem, Reston, VA, USA), or vehicle control (0.01% DMSO). The concentration of the antagonists was selected based on recommendations from the company and prior published studies [24,31,32,33].

### 2.5. Chemotaxis Assay

PBMCs were isolated from blood of OAC patients by density gradient centrifugation and activated for 72 h in RPMI 1640 medium (Gibco, Waltham, MA, USA) supplemented with 10% foetal bovine serum and penicillin–streptomycin (Gibco, Waltham, MA, USA) using plate-bound anti-CD3 (10 μg/mL, Biolegend, San Diego, CA, USA), anti-CD28 (10 μg/mL, Ancell, Bayport, MN, USA) and recombinant human IL-2 (10 IU/mL, Immunotools, Friesoythe, Germany) for 3 days. PBMCs were irradiated with a 1.8 Gy dose of irradiation on day 1 and day 2, 24 h apart, or cells were non-irradiated (NIR). On day 3, PBMCs were resuspended in serum-free RPMI 1640 medium and treated with 80 μM of Maraviroc (Axon MedChem, Reston, VA, USA) or 245 nM of CX_3_CR1 antagonist AZD8798 (Axon MedChem, Reston, VA, USA) for 1 h. Cells were subsequently added at a density of 0.2 × 10^6^ cells/100 µL serum-free RPMI 1640 medium to a 5 µm pore Transwell filter system (Corning, Corning, NY, USA) with TCM or IR-TCM added in the lower chamber. M199 alone was used as a negative control, and M199 supplemented with 20% FBS was used as a positive control. This system was incubated for 2 h at 37 °C, 5% CO_2_. Cells were collected from the lower chamber and stained for flow cytometric analysis with CD3-PerCP, CD4-BV510 (Biolegend, San Diego, CA, USA), and CD8-BV421 (BD Biosciences, Franklin Lakes, NJ, USA). CountBright beads (Thermo Fisher Scientific, Waltham, MA, USA) were used to enumerate the migrated CD3^+^CD4^+^ cells and CD3^+^CD8^+^ cells. Cells were acquired using BD FACS CANTO II (BD Biosciences) and analysed using FlowJo software v10 (Tree Star, Ashland, OR, USA).

### 2.6. Flow Cytometry

PBMCs were stained with zombie aqua viability (Biolegend, San Diego, CA, USA) dye. Antibodies used for staining included CCR5-PE, CX_3_CR1-AF647, CCR1-APC, CD3-PerCP, CD4-BV510 (Biolegend, San Diego, CA, USA), CD8-BV421 (BD Biosciences, Franklin Lakes, NJ, USA). PBMCs were resuspended in FACS buffer and acquired using BD FACS CANTO II (BD Biosciences Franklin Lakes, NJ, USA) using Diva software version 8 and analysed using FlowJo v10 software (TreeStar Inc., Ashland, OR, USA). For intracellular cytokine staining, PBMCs were treated with PMA (10 ng/mL) and ionomycin (1 µg/mL) for the last 4 h of the anti-CD3/28 activation. CD107a-PE (BD Biosciences, Franklin Lakes, NJ, USA) was also added to the cells for the last 4 h of the activation process. For the last 3 h of the activation, PBMCs were treated with brefeldin A (10 µg/mL, eBiosciences, San Diego, CA, USA). Cells were harvested and washed in FACS buffer, and intracellular cytokines were assessed using a Fixation/Permeabilisation kit (BD Biosciences, Franklin Lakes, NJ, USA), as per manufacturer’s recommendations. Cells were stained with cell surface antibodies (CD8-BV421, CD3-PerCP, and CD4-APC (Biolegend, San Diego, CA, USA)) washed, permeabilised, and then stained with IFN-γ-BV510 (Biolegend, San Diego, CA, USA). Cells were resuspended in FACS buffer and acquired using BD FACS CANTO II (BD Biosciences, Franklin Lakes, NJ, USA).

### 2.7. Statistical Analysis

Data were analysed using GraphPad Prism version 10 (GraphPad Prism, San Diego, CA, USA) software and were expressed as mean ± SEM. Statistical differences between treatments within cancer donors or within healthy donors were analysed using paired non-parametric *t*-test, and statistical differences between treatments between healthy donors and cancer donors were analysed using unpaired non-parametric *t*-tests. Statistical significance was determined as *p* ≤ 0.05.

## 3. Results

### 3.1. Clinically Relevant Doses of Radiation Ex Vivo Can Rescue Diminished Effector T Cell Function in OAC Patients

The suppression of anti-tumour T cell immunity is commonly observed across a wide spectrum of tumour types [34]. Therapies such as radiotherapy emerged as useful tools to boost anti-tumour T cell immunity and propagate pre-existing anti-tumour immune responses. In this study, we profile anti-tumour T cell phenotypes in OAC patients and age-matched healthy donors as a comparison and investigate if the use of clinically relevant doses of irradiation might promote anti-tumour T cell profiles (Figure 1a). IFN-γ is a key T cell cytokine that directly induces tumour cell death and promotes the anti-tumour effector function of other immune cells. We observed that CD4^+^ T cells expanded from the peripheral blood of OAC patients produced significantly less IFN-γ compared with healthy T cell controls (healthy donors: 29.82 ± 6.0% vs. OAC donors: 14.08 ± 2.1%, *p* = 0.04) (Figure 1b,c). There was no significant difference in IFN-γ production in the CD8^+^ T cell compartment between OAC donors and healthy donors (Figure 1b).

CD8^+^ T cells secrete lytic granules to kill target cancer cells, a process which involves the fusion of the granule membrane with the cytoplasmic membrane of the T cell, resulting in surface exposure of lysosomal-associated proteins that are typically present on the lipid bilayer surrounding lytic granules, such as CD107a [35]. Our data showed that the cytotoxic potential of expanded CD8^+^ T cells is significantly lower in OAC patients compared with healthy controls, indicated by a significantly lower frequency of CD107a^+^ CD8^+^ T cells [33] (healthy donors: 26.37 ± 2.2% vs. OAC donors: 9.39 ± 1.9%, *p* = 0.002) (Figure 1d,e).

We found that clinically relevant doses of irradiation enhanced the diminished T cell effector function in OAC patients. Irradiation significantly increased IFN-γ production in expanded viable CD4^+^ T cells from OAC patients (NIR: 14.08 ± 2.1% vs. IR: 36.61 ± 8.6%, *p* < 0.01, (Figure 1g,h)) and significantly increased cytotoxic degranulation by viable CD8^+^ T cells (NIR: 9.39 ± 1.9% vs. IR: 19.8 ± 3.8%, *p* = 0.03, (Figure 1j,k)). Irradiation had no effect on the production of IFN-γ by viable CD4^+^ or CD8^+^ T cells or cytotoxic degranulation by CD8^+^ T cells derived from healthy donors (Figure 1f,i).

Collectively, these data reveal that the expansion of circulating T cells from OAC patients has a diminished effector function, which can be rescued by administering clinically relevant doses of irradiation to T cells ex vivo.

### 3.2. Significantly Higher Frequencies of Circulating CCR5^+^ T Cells in OAC Patients Compared with Healthy Controls, While CCR5 Surface Expression Is Upregulated by Clinically Relevant Doses of Radiation

Chemokines are critical for directing immune cells towards the solid TME, while their broader effects on immune cell phenotype and function are still being established [24,36]. In the context of OAC, CCR1, CCR5, and CX_3_CR1 were identified as key drivers of erroneous T cell migration to the omentum at the expense of effective anti-tumour immunity [24,27,36]. Furthermore, the antagonism of these chemokine receptors was shown as a viable approach to redirect T and NK cells away from the omentum [31,32]. Moreover, while CCR5 and CCR1 ligands have been detected in abundance in the OAC TME, T cells expressing their cognate receptors are not detected in equal abundance [19]. Therefore, elucidating their potential in the context of boosting T cell infiltration of OAC tumours and T cell function in combination with irradiation is warranted. Regulatory T cells utilize the chemokine receptor CCR5 to mobilize to tissue-specific sites [27]. In this study, we profiled chemokine receptor expression on T cells from OAC patients and used age-matched healthy controls as a comparison (Figure 2a). The expression of CCR1, CCR5, and CX_3_CR1 on T cells was investigated following a 2-day T cell activation period. The surface expression of CCR1 and CX_3_CR1 was comparable between T cells from OAC donors and healthy donors (Figure 2b,e). However, the surface expression of CCR5 was significantly higher on OAC-derived CD4^+^ and CD8^+^ cells compared with healthy donors (CD4^+^: healthy donors: 4.89 ± 1.0% vs. OAC donors: 26.80 ± 5.4%, *p* = 0.01, CD8^+^: healthy donors: 5.82 ± 1.1% vs. OAC donors: 32.01 ± 5.5%, *p* < 0.01) (Figure 2c,d).

We also assessed the effect of clinically relevant doses of irradiation on chemokine receptor expression profiles (Figure 2f,g). We found that irradiation substantially increased CCR5 expression on the surface of CD8^+^ T cells from OAC patients but not healthy donors (NIR: 32.0 ± 5.5% vs. IR: 37.3 ± 6.4%, *p* = 0.05) (Figure 2h,i). Irradiation did not significantly affect the expression of CCR1 or CX_3_CR1 on T cells from healthy donors or OAC donors.

Overall, these findings revealed that the expression of CCR5 was significantly elevated on OAC patient-derived T cells compared with healthy donors and that clinically relevant doses of irradiation substantially increased the expression of CCR5 on CD8^+^ T cells from OAC patients.

### 3.3. Irradiation Increased the Migratory Capacity of OAC-Derived T Cells toward OAC Patient-Derived Tumour-Conditioned Media

We have shown that clinically relevant doses of radiation can alter the chemokine receptor expression profiles of T cells. These alterations might have either positive or detrimental implications for the T cell migration towards the tumour compartment. Recognizing that radiotherapy forms the current standard of care for OAC patients and that it is emerging as a valuable tool to boost anti-tumour immunity, it is important to understand if clinically relevant doses of radiation might affect T cell migration towards the tumour compartment or irradiated tumour compartment (Figure 3a). OAC tumour biopsies were non-irradiated (NIR) or irradiated with 1.8 Gy (IR), and the non-irradiated tumour-conditioned media (TCM) and irradiated TCM (IR-TCM) were harvested following 24 h. The generated TCM and IR-TCM recapitulate the chemokine profile of the OAC tumour basally and following a clinically relevant dose of irradiation (Figure 3a). Firstly, we tested and proved that the OAC-derived non-irradiated (Figure 3b) and irradiated T cells (Figure 3c) could respond to the chemotactic cues of 20% FBS and migrate across the transwell. Next, we investigated if irradiated T cells altered their migratory capacity toward the NIR-TCM or the IR-TCM. We observed an increase in the migration of irradiated CD4^+^ T cells towards M199 and TCM compared with non-irradiated CD4^+^ T cells, suggesting that irradiation equipped CD4^+^ T cells with an enhanced migratory capacity (CD4^+^ T cells—non-IR: 1.19 ± 0.4 vs. IR: 2.38 ± 0.5-fold change, *p* = 0.03), (Figure 3d). In addition, we also observed an increase in the migration of irradiated CD8^+^ T cells towards M199, but this was not parallel with an increase in the migration of CD8^+^ T cells toward TCM compared with non-irradiated CD8^+^ T cells, suggesting that irradiation equipped CD8^+^ T cells with an enhanced migratory capacity but not with specific chemotactic queues to increase their migration toward TCM (Figure 3e). However, irradiating CD4^+^ and CD8^+^ T cells did not significantly increase the migration of T cells toward the irradiated TCM. Surprisingly, we noticed that irradiated CD8^+^ cells migrated significantly less toward irradiated TCM compared with non-irradiated TCM (3.60 ± 1.0 vs. 1.38 ± 0.3-fold change, *p* = 0.05) (Figure 3e).

The increase in the migration of irradiated CD4^+^ T cells towards the TCM suggests that the irradiation equipped T cells with an enhanced migratory capacity that might be exploited with therapeutic intent.

### 3.4. In Vitro Treatment with CCR5 Antagonist Maraviroc Increased the Migration of Irradiated CD8^+^ T Cells towards the Irradiated Tumour Compartment and Enhanced Production of IFN-γ by CD4^+^ T Helper Cells

In light of previous findings in OAC patients demonstrating that CCR1 and CX_3_CR1 signalling in T cells led to the erroneous migration of T cells away from the tumour and toward the omentum and liver, as well as findings detailing how their antagonism enhanced their migration to the OAC TME [19,24,31,32,37], we sought to investigate if CCR5 antagonism might affect the migration of T cells to the OAC TME. CCR5 is expressed on effector T cells and recruits effector T cells to the tumour; however, CCR5 is also expressed on regulatory T cells and has been implicated in recruiting regulatory T cells to the tumour microenvironment in colorectal cancer [27]. Considering our findings indicate that CCR5 is increased on the surface of OAC-derived T cells and that clinically relevant doses of irradiation further upregulated CCR5 expression, this pathway may represent a therapeutically exploitable immunomodulatory pathway in OAC. Therefore, we assessed if CCR5 antagonism could alter the migration of non-irradiated or irradiated OAC patient-derived T cells toward the non-irradiated or irradiated OAC TME (TCM) using a Transwell assay (Figure 4a). We observed that CCR5 antagonism did not affect the number of CD4^+^ T cells migrating toward the OAC TCM in the absence or presence of irradiation (Figure 4b). However, CCR5 antagonism significantly increased the frequency of irradiated CD8^+^ T cells migrating towards the irradiated TCM (untreated: 1.38 ± 0.3 vs. CCR5 antagonist: 2.27 ± 0.6, *p* = 0.03) (Figure 4c).

Given the recently reported effects of “driver” chemokines on T cell function, we next investigated if CCR5 antagonism could enhance anti-tumour effector function in OAC patient blood-derived T cells. To test this, CCR5 signalling in T cells was activated via treatment with its cognate ligand MIP-1α for 48 h and CCR5 signalling was antagonized by treating with a CCR5 receptor antagonist (Maraviroc) in combination with the activating MIP-1α chemokine ligand. These experiments were conducted in the absence and presence of clinically relevant doses of irradiation (Figure 5a). Similar experiments were conducted to evaluate the effect of CCR1 and CX_3_CR1 antagonism on T cell function (Figure 5a). The production of IFN-γ or the cytotoxic potential of T cells was not significantly affected by CCR1 antagonism and CX_3_CR1 antagonism (Figure 5b,d,g). We observed that antagonizing CCR5 signalling significantly increased IFN-γ production in non-irradiated OAC-derived CD4^+^ T cells (MIP-1α: 15.10 ± 5.0% vs. MIP-1α + Maraviroc: 29.92 ± 4.4%, *p* = 0.04) (Figure 5c,e,f). The cytotoxic potential of T cells was not significantly affected by CCR5 antagonism (Figure 5g).

Although it does appear that there is a trend toward a decrease in IFN-γ production by CD4^+^ and CD8^+^ cells and CD107a degranulation by CD8^+^ cells upon treatment with Maraviroc or MIP-1α in combination with irradiation treatment specifically, this trend does not reach statistical significance (Figure 5c,g and Appendix A). Paradoxically, a blockade of the CCR5 pathway in the absence of irradiation significantly increases the production of IFN-γ by CD4^+^ cells. It is difficult to speculate why the activation or blockading of the CCR5 pathway in T cells in the presence of irradiation may somewhat decrease IFN-γ production or cytotoxic potential but have the opposite effect in the absence of irradiation. Further studies will be required to understand the confounding interaction between irradiation and the CCR5-MIP-1α axes in T cells.

## 4. Discussion

Radiation can elicit potent anti-tumour immune responses by working in tandem with the immune system. Matsumura et al. [38] demonstrated that radiation stimulates tumour cells to release chemokines such as CXCL16 to recruit effector T cells to the tumour. Other favourable effects mediated by radiation include the release of tumour antigens and damage-associated molecular patterns by tumour cells leading to an increase in antigen presentation to T cells [39]. Collectively, this triggers anti-tumour immunity and tumour infiltration by lymphocytes, facilitating the immune-mediated clearance of tumour cells [40,41]. Radiation also enhances a pro-inflammatory environment via the activation of the STING pathway [41]. These combined effects may theoretically contribute to the remodelling and reprogramming of the TME to transform “cold” tumours with less immune infiltrate into “hot” tumours, enriched with an immune infiltrate that is conducive to a favourable response to immunotherapy [42,43]. In this study, we first profiled the anti-tumour function of circulating T cells in OAC patients and compared them with healthy donor controls to assess the level of systemic immunosuppression in OAC patients. Using ex vivo models, we then tested if clinically relevant doses of irradiation could alleviate systemic immunosuppression in circulating T cells, thus evaluating the therapeutic utility of radiation as an immunostimulatory agent in OAC patients. Given that CCR5 was significantly elevated on circulating T cells derived from OAC patients, we then focussed our study on elucidating the effects of CCR5 antagonism on anti-tumour T cell function and migration of T cells toward the OAC compartment in the absence and presence of radiation.

Our findings reinforce our previous data highlighting dysregulated immune responses in OAC patients [12], whereby we observed that effector T cell function is diminished in OAC patients but could be rescued with clinically relevant doses of radiation. These results support the established role of radiotherapy as an immunostimulatory approach to boost anti-tumour immunity [44]. Intriguingly, irradiating T cells with clinically relevant doses of radiation increased their migratory capacity, directing them toward chemotactic cues present in the OAC TME. This result supports previous findings indicating that radiation therapy can enhance the recruitment of T cells to the solid TME [45]. Circulating T cells are exposed to radiation when they pass through the irradiation treatment field; typically, ~2 whole blood volumes circulate through the tumour irradiation field during each radiotherapy treatment [22]. In addition, to minimize local tumour recurrence, the radiation field is also tailored to target the tumour-draining lymph nodes as well as the tumour, which exposes the lymph node residing T cells to irradiation [46].

Despite radiation having attractive qualities in promoting anti-tumour immunity, inevitably, immunosuppressive mechanisms are also enhanced by radiation [47,48]. These paradoxical effects of radiation on immuno-stimulation or immuno-suppression are observed in different scenarios and are likely context-dependent but may contribute to the success or failure of radiation and combination radiotherapy approaches [47]. Previous work by our group identified abundant levels of CCR5 ligands RANTES and MIP-1α in the OAC TME but did not detect an equal abundance of T cells expressing the receptor [19]. Moreover, this study uncovered an impaired migratory capacity of OAC patient-derived T cells, which we propose might compromise effective T cell infiltration of tumour and anti-tumour immunity in OAC patients [19]. A study by Oliveira et al. pinpointed an immunosuppressive role for CCR5 in the TME via the recruitment of regulatory T cells, which facilitated tumour development and progression in squamous cell skin carcinoma [49]. Tregs with higher CCR5 expression were more immunosuppressive than Tregs expressing lower levels of CCR5 expression [33]. However, antagonizing CCR5 did not inhibit the recruitment of regulatory T cells to the TME in murine models of colorectal cancer, but CCR5 antagonism did delay tumour growth [33]. CCR5 is upregulated in many tumours and is often a poor prognostic indicator [50,51]. Other reports from our group showed that CCR5 ligands are enriched in extratumoural compartments of the omentum and liver of OAC patients and that CCR1 and CCR5 antagonism can limit erroneous migration of T cells towards these tissues [32]. Therefore, these pathways influenced T cell migratory pathways in OAC patients and were interrogated here. We observed that CCR5^+^ T cells were significantly more prevalent in the blood of OAC patients compared with healthy controls. Given the emergence of chemokines as “drivers” of T cell polarisation and function, we examined the effects of the CCR5 pathway on the Th1 cytokine profile and migration of OAC patient-derived T cells.

In our study, CCR5 antagonism enhanced anti-tumour effector T cell function by increasing IFN-γ production by CD4^+^ T cells, which plays an important role in mediating tumour regression. Furthermore, CCR5 antagonism increased the migration of irradiated CD8^+^ T cells toward the irradiated TME compartment. Complementary studies in pre-clinical models demonstrated that CCR5 antagonists show promise as anti-cancer therapies and have been recognized as a potential therapeutic target for cancer [51]. In the setting of OAC, these data, together with our previous reports, suggest that CCR5 may limit the inappropriate migration of T cells to extratumoural tissues while enhancing their anti-tumour capabilities.

Within the expanded lymphocyte population, there likely exists a mixed population of both pro- and anti-tumour T cells, and this is in keeping with our previous data [52]. Further studies will fully interrogate their specific immune phenotype and whether this can translate into an improvement in tumour control in murine models.

Previous reports demonstrated that the CCR5/CCL5 axis promotes a migratory and invasive phenotype in pancreatic cancer cells [53] and Hodgkin’s lymphoma [54] via tumour cell-intrinsic signalling of CCR5 on tumour cells. Collectively, these studies demonstrate that the CCR5 axis also possesses immune-independent functions in promoting metastasis, a key hallmark of cancer apart from its more well-known role in immune regulation. These studies underscore the therapeutic potential of blocking CCR5 for inhibiting metastasis as well as enhancing anti-tumour immunity. Given the redundancy that exists across the chemokine network, it is quite likely that the CCR5-MIP-1α interaction may not be essential for the enhancement of CD4^+^ and CD8^+^ anti-tumour responses. Redundancy across cancer-promoting pathways and immunoregulatory pathways is a constant and significant barrier to successful cancer therapy and must be considered when designing new therapeutic combinations to circumvent the emergence of resistance mechanisms. Further studies on the role of CCR5-MIP-1α in enhancing anti-tumour T cell immunity are warranted, as well as the assessment as to whether dual chemokine receptor antagonism would be preferable.

Surprisingly, we did observe that irradiated CD8^+^ T cells migrated significantly less toward the irradiated TCM compared to the non-irradiated TCM. Further investigation as part of future studies will be required to elucidate the precise mechanism of action mediating this effect. It is known from other cancer types that irradiation markedly alters the tumour microenvironment, increasing the production of pro-inflammatory cytokines and secreting T cell recruiting chemokines [2,4]. However, in this specific context and ex vivo model of OAC, it appears that irradiation is having profound effects on the OAC tumour microenvironment that are decreasing the recruitment of CD8^+^ T cells. A potential explanation could be that irradiation may have altered the chemokine secretome within the tumour microenvironment of these OAC explant models, which may not support the recruitment of CD8^+^ T cells. The cytostatic/cytotoxic effects of irradiation on both immune cells and tumour cells may result in a decrease in the production of the chemokines responsible for recruiting CD8^+^ cells as well. Additional studies using different models of OAC will be necessary to determine the factors governing the altered migratory pattern of irradiated T cells to non-irradiated versus irradiated TCM. Although these tumour explant models offer valuable insight, they also carry their own set of limitations, which include lack of a vascular system, extracellular matrix, and tumour-draining lymph nodes, all of which elicit their own chemotactic cues and ultimately impact immune cell decision-making and trafficking. In addition, the secretions from the irradiated OAC tumour explants (TCM) were collected 24 h after irradiation, and perhaps at longer timepoints, we might observe different effects. In vivo studies will be important for future interrogation and validation of this work and to deepen our mechanistic understanding of the impact of the CCR5-MIP-1α axis on T cell infiltration of OAC tumours.

In conclusion, the findings from this study underline the important immunostimulatory role of radiotherapy in OAC to enhance anti-tumour effector T cell function and promote T cell infiltration of OAC tumours. The revelation of CCR5 antagonism as a synergistic approach to combine with radiotherapy to enhance anti-tumour effector T cell function is a compelling therapeutic concept and warrants further interrogation of the repurposing of the FDA-approved CCR5 antagonist Maraviroc for the treatment of OAC.

## Figures and Tables

**Figure 1 biomedicines-12-00819-f001:**
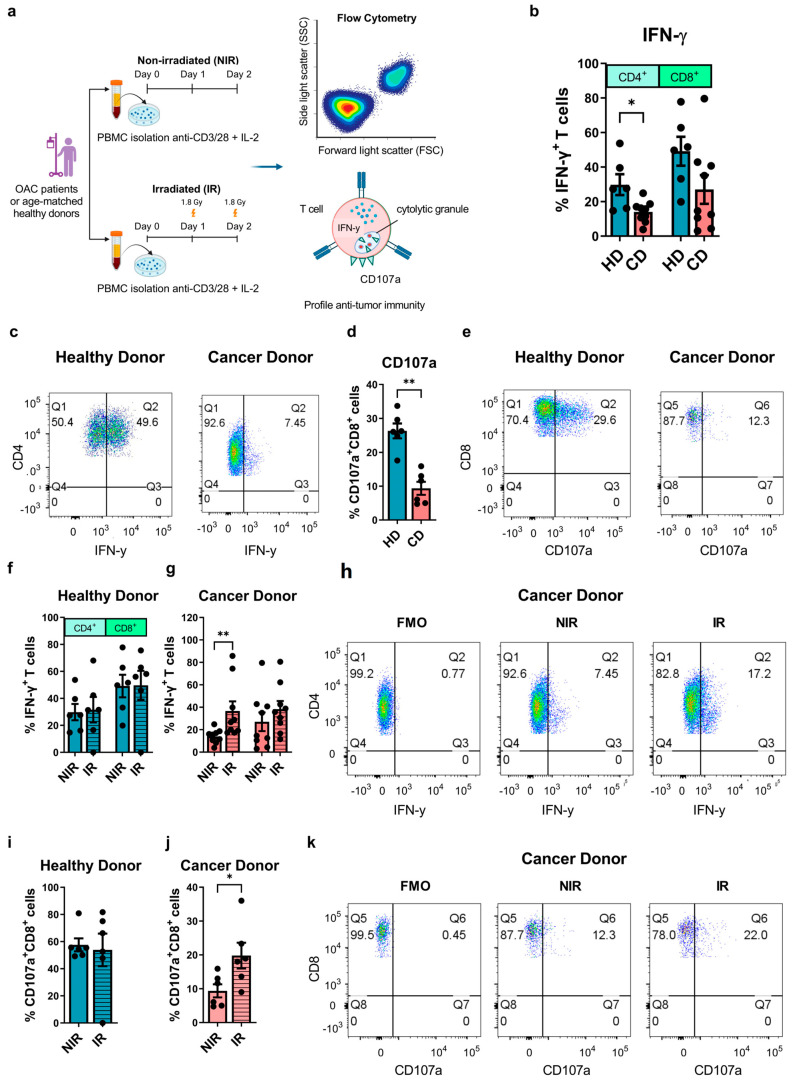
Effector T cell functions were diminished in OAC patients compared with healthy controls and were enhanced using clinically relevant doses of irradiation ex vivo. (**a**) PBMCs isolated from age-matched healthy donors (HD) (*n* = 6) and treatment−naïve OAC cancer donors (CD) (*n* = 9) were activated with plate−bound anti−CD3 and anti−CD28 agonists for 72 h receiving 2 × 1.8 Gy fractions of irradiation (irradiated, IR) on day 1 and day 2, 24 h apart, or were non-irradiated, NIR. The frequency of viable T cells producing IFN-γ and expressing CD107a was then assessed by flow cytometry in HDs (**b**,**c**) and CDs (**d**,**e**). The effect of IR on IFN-γ production by viable T cells in HDs (**f**) and CDs (**g**,**h**) is also shown. The effect of IR on CD107a expression on CD8^+^ T cells in HDs and CDs is depicted in (**i**) and (**j**,**k**), respectively. Mann−Whitney tests were used to compare between HD and CD groups. Wilcoxon signed−rank tests were used to compare the effect of NIR with IR in the same donors. All analyses were conducted on viable T cells using a zombie dye to exclude dead cells, and fluorescence minus-one controls (FMO) were used for gating analysis. * *p* < 0.05, ** *p* < 0.01.

**Figure 2 biomedicines-12-00819-f002:**
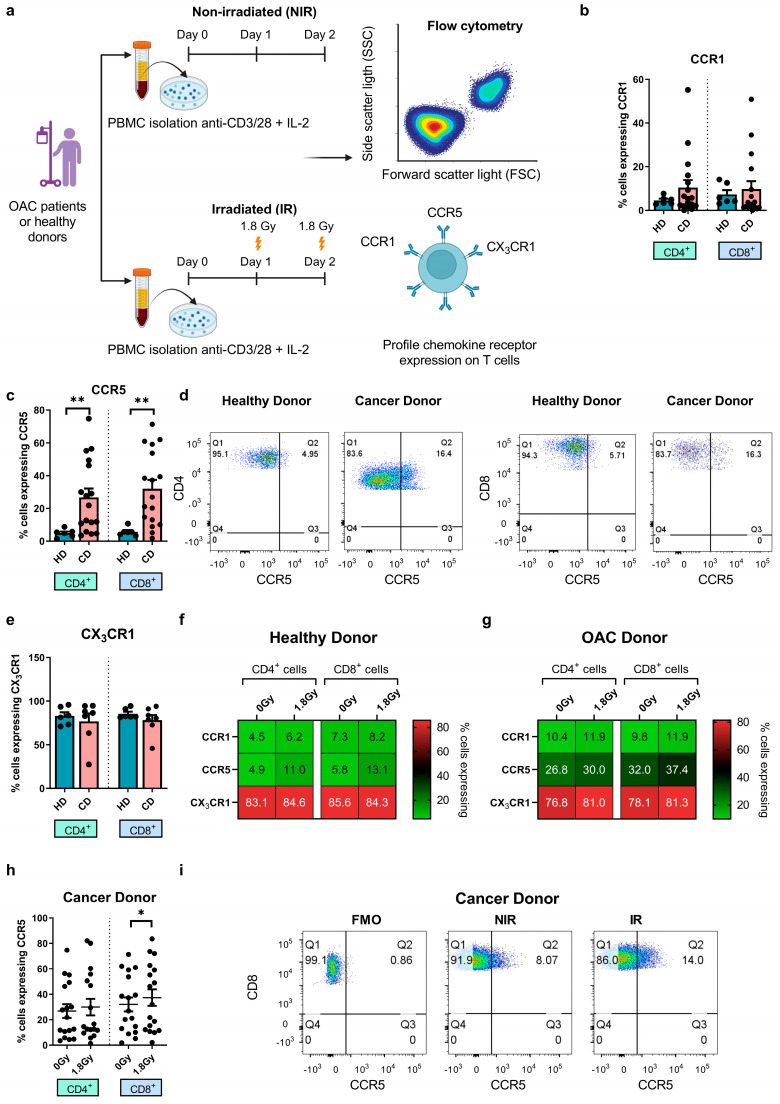
Expanded circulating T cells from OAC patients expressed higher levels of CCR5, which was further upregulated on the surface of CD8^+^ T cells by clinically relevant doses of irradiation. (**a**) PBMCs isolated from treatment−naïve OAC donors (*n* = 7) and age—matched HDs (*n* = 6) were activated with plate−bound anti−CD3 and anti−CD28 agonists for 72 h, receiving 2 × 1.8 Gy fractions of irradiation (irradiated, IR) on day 1 and day 2, 24 h apart, or were non−irradiated, NIR. The frequencies of T cells in HDs and CDs expressing CCR1 (**b**), CCR5 (**c**,**d**), and CX_3_CR1 (**e**) were assessed by flow cytometry. The effect of IR on CCR1, CCR5, and CX_3_CR1 expression on T cells from HDs (**f**) and CDs (**g**) was also assessed by flow cytometry and depicted in heat maps shown (% cells expressing). The effect of IR on CCR5 expression on T cells from CDs is displayed in (**h**,**i**). All analyses were conducted on viable T cells using a zombie dye to exclude dead cells, and fluorescence minus-one controls (FMO) were used for gating analysis. Mann−Whitney tests were used to compare between HD and CD groups. Wilcoxon signed-rank tests were used to compare the effect of NIR with IR in the same donors. * *p* < 0.05, ** *p* < 0.01.

**Figure 3 biomedicines-12-00819-f003:**
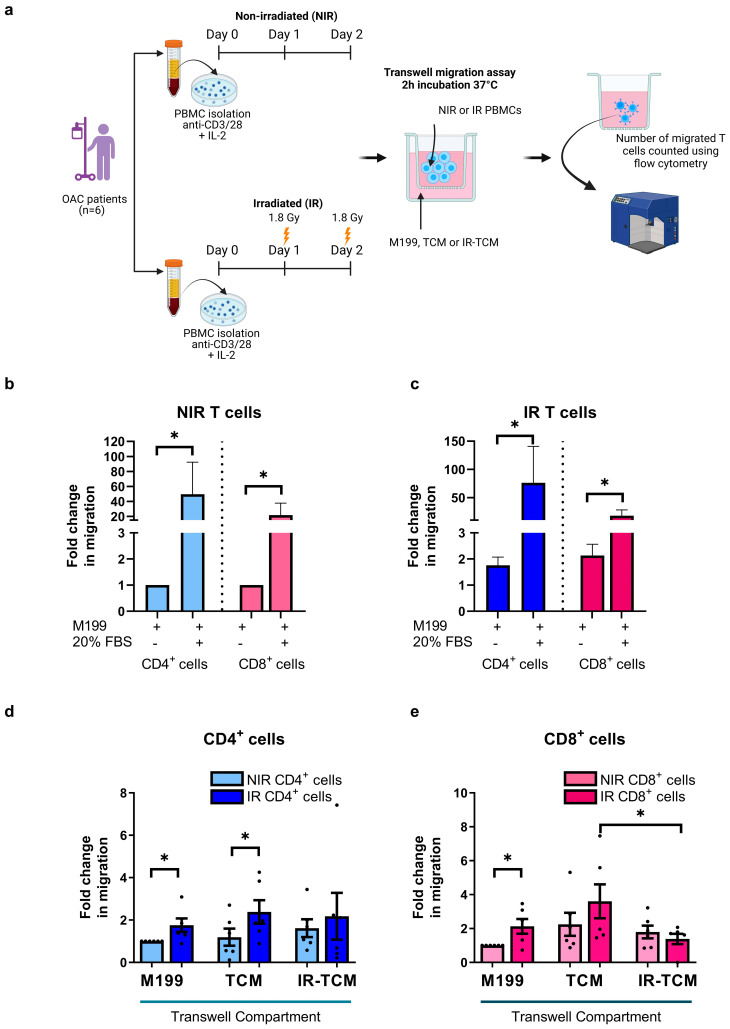
Clinically relevant doses of irradiation ex vivo significantly increased the migration of OAC patient−derived T cells toward tumour-conditioned media. (**a**) PBMCs isolated from treatment−naïve OAC donors (*n* = 6) were activated with plate-bound anti−CD3 and anti−CD28 agonists for 72 h, receiving a 1.8 Gy fraction of irradiation on day 1 and day 2 or were non-irradiated (NIR). The number of migrating CD4^+^ and CD8^+^ cells to control media (M199), TCM, and irradiated TCM (IR-TCM) was assessed by flow cytometry and expressed as fold changes relative to the NIR cells migrating towards the M199 control. (**b**,**c**) demonstrate that NIR and IR T cells can respond to chemotactic queues and migrate towards a 20% FBS control, respectively. (**d**,**e**) depict the fold-change in migration of NIR and IR CD4^+^ and CD8^+^ cells towards an M199 control, TCM, and IR-TCM, respectively. All analyses were conducted on viable T cells using a zombie dye to exclude dead cells. Mann–Whitney tests were used to compare the migration of T cells towards the different Transwell compartments. * *p* < 0.05.

**Figure 4 biomedicines-12-00819-f004:**
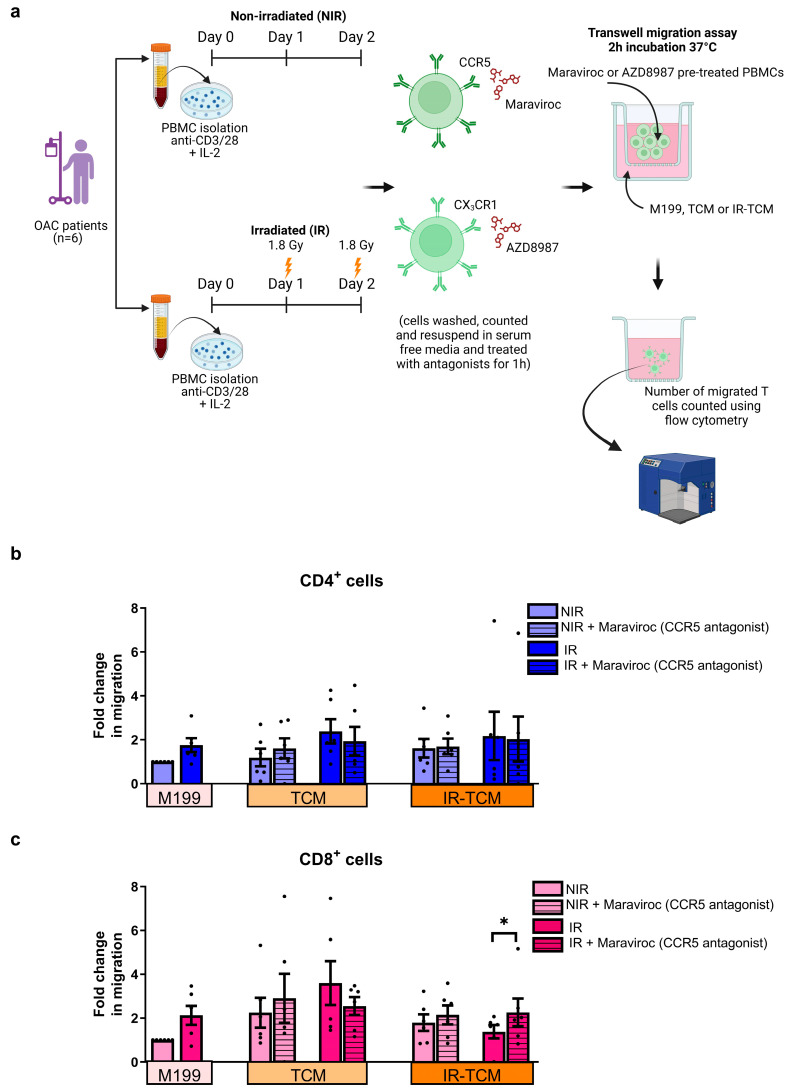
CCR5 antagonism with Maraviroc increased the migration of the number of irradiated CD8^+^ T cells towards irradiated OAC tumour-conditioned media. (**a**) PBMCs isolated from treatment−naïve OAC donors (*n* = 6) were activated with anti−CD3 and anti−CD28 agonists for 72 h, receiving a 1.8 Gy fraction of irradiation on days 1 and 2 or were non-irradiated (NIR). PBMCs were then treated with Maraviroc (CCR5 antagonist) for one hour. Using a Transwell migration assay, the number of migrating CD4^+^ (**b**) and CD8^+^ T (**c**) cells to control M199 media, TCM, and irradiated (IR) TCM was then assessed using counting beads by flow cytometry and expressed as fold changes relative to the M199 control. All analyses were conducted on viable T cells using a zombie dye to exclude dead cells. Mann–Whitney statistical tests were used to compare the migration of T cells towards the different Transwell compartments. * *p* < 0.05.

**Figure 5 biomedicines-12-00819-f005:**
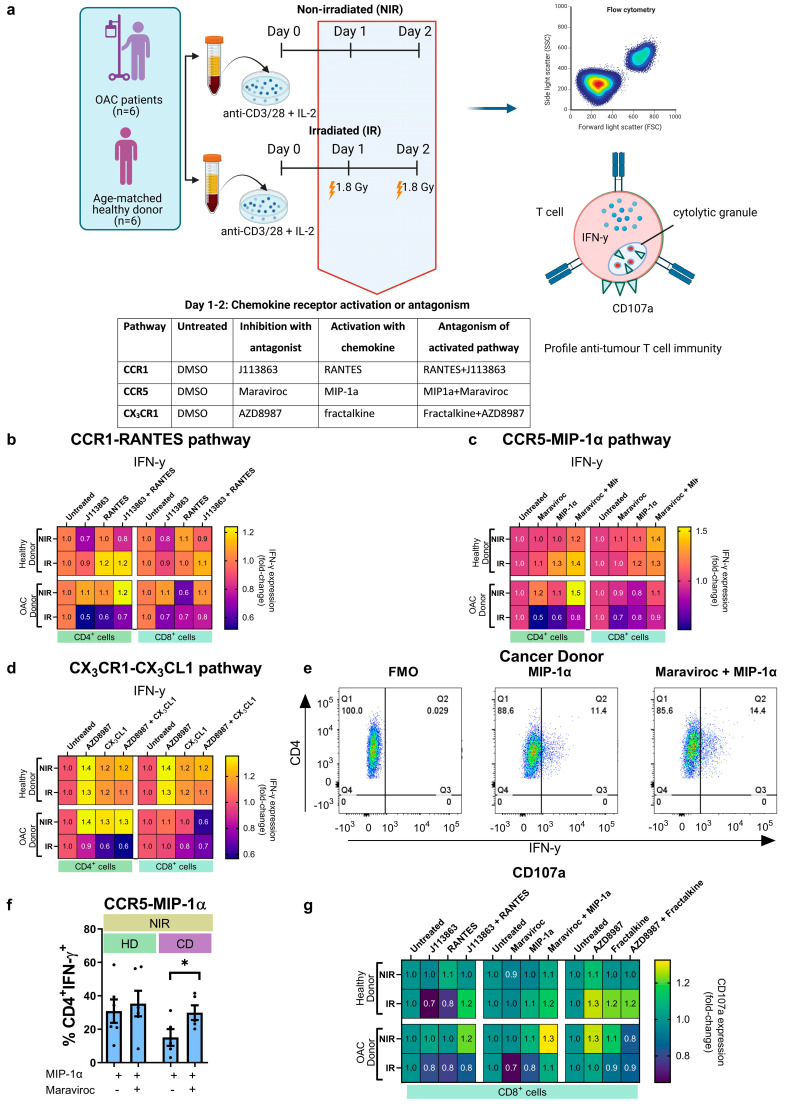
CCR5 antagonism significantly increased IFN-γ production by OAC patient−derived CD4^+^ T helper cells. (**a**) PBMCs isolated from treatment-naïve OAC donors (*n* = 5) and age−matched non-cancer donors (*n* = 6) were activated with anti-CD3 and anti-CD28 agonists for 72 h and treated with J113863 (CCR1 antagonist) and RANTES, AZD8987 (CX_3_CR1 antagonist) and fractalkine or Maraviroc (CCR5 antagonist) and MIP-1α. The PBMCs also received 2 × 1.8 Gy fractions of irradiation on day 1 and day 2, 24 h apart, or were non-irradiated (NIR). The frequency of cells producing IFN-γ and CD107a was assessed by intracellular and extracellular flow cytometry, respectively. (**b**–**d**) heat maps depicting the effect of activating or antagonising the CCR1−RANTES, CCR5-MIP-1α, and CX_3_CR1−fractalkine pathways on the percentage of CD4^+^ and CD8^+^ T cells producing IFN-γ, respectively, relative to the untreated control. (**e**) Graphical display showing the effect of CCR5 antagonism on the production of IFN-γ by T cells, with representative dot plots shown in (**f**). (**g**) The effect of activating or antagonizing CCR1-RANTES, CCR5-MIP-1α, and CX_3_CR1-fractalkine pathway on CD107a degranulation by CD8^+^ T cells. All analyses were conducted on viable T cells using a zombie dye to exclude dead cells, and FMO controls were used for gating analysis. Paired parametric *t*-test was used to compare between two groups * *p* < 0.05.

**Table 1 biomedicines-12-00819-t001:** Patient demographic table.

	Cancer Cohort for Blood Samples	Cancer Cohort for Tumour Samples	Healthy Donor Cohort for Blood Samples
Patient Demographic Table	*n* = 9	*n* = 6	*n* = 6
Age (years)	(51–75) 63.2	(48–75) 61.0	(55–61) 57.8
Sex ratio (M:F)	7:2	5:1	5:1
Diagnosis (no. patients)	OAC (*n* = 9)	OAC (*n* = 6)	Non-cancer (*n* = 6)
Clinical tumour stage (no. patients)			
T0	0	0	
T1	1	0	
T2	2	2	
T3	6	4	
T4	0	0	
Clinical nodal status (no. patients)			
Negative	4	2	
Positive	5	4	

## Data Availability

Data can be requested by contacting the corresponding author of this study.

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
