# Peer review of "Analysing the Combined Effects of Radiotherapy and Chemokine Receptor 5 Antagonism: Complementary Approaches to Promote T Cell Function and Migration in Oesophageal Adenocarcinoma"

_biomedicines, 2024, doi:10.3390/biomedicines12040819_

Round 1

Reviewer 1 Report

Comments and Suggestions for Authors

The authors investigated the immunomodulatory effects of RT and chemokine receptor antagonists on the T-cell phenotype of patients with OAC. The immunostimulatory properties of radiation in promoting anti-tumor T cell responses and increasing T cell migration toward chemotactic cues in the tumor were also highlighted.

 The authors conducted diverse studies using tumor tissue and lymphocytes from patients with PAC to demonstrate the effects of RT on the migratory and antitumor potential of CD4+ T cells and CD8+ T cells. However, their experimental results are not necessarily consistent with their conclusions. There are some points that should be reconsidered.

The title is “Synergistic effects of radiotherapy of radiotherapy and CCR5 antagonism on T cell function and migration in oesophageal adenocarcinoma”. Do RT and the CCR5 antagonist act synergistically? Has this been demonstrated?

It is unknown what factors are released from tumor tissue or irradiated tumor tissue. Are those factors reduced by RT? Explanation needed.

Reference 25 reports that CCR5 is highly expressed in Tregs infiltrating human colon cancer. Do the CD4+ T cells used in this study not contain CD4+ Tregs?

Figures 3 and 4: IR appears to increase CD4+ and CD8+ T cell motility. In experiments with TCM, maraviroc reduced, although not significantly, the migration of CD4+ and CD8+ T cells (Figure 4b, c). This is a reasonable result based on its inhibitory effect on the CCL5-CCR5 pathway of T cells. However, migration of IR-CD8+ T cells to IR-TCM was markedly inhibited and partially restored by maraviroc. The authors selected the results of their experiments with IR-TCM and concluded that maraviroc increased migration of CD8+ T cells to chemotactic signals in the tumor. It is important to explain the reason for this difference in migration between TCM and RI-TCM. Why is there less migration of IR-CD8+ T cells to IR-TCM (Figure 4c)? Furthermore, it is also important to clarify whether the CCR5 agonist MIP-1a really promotes the migration of CD4+ and CD8+ T cells.

Figure 5: In Figure 5e, the authors selected the NIR-CD4+ results in Figure 5c and stated that the CCR5 antagonist maraviroc enhanced IFN-g production. However, in the CCR5-MIP-1a pathway, the most striking change is the reduction of IFN-g production/CD107a in IR-CD4+ T cells or CD8+ T cells of OAC patients treated with maraviroc or MIP-1a. The authors need to explain these events. Otherwise, the conclusion that maraviroc enhanced IFN-g production by CD4+ is not convincing.

Previous studies have shown that CCR5/CCL5 interaction promotes migration and invasiveness of cancer cells and lymphomas. There, CCR5 expressed on tumor cells has been shown to play a major role (e.g., Liu-J et al. Oncotarget 2015, 6, 24978; Casagrande-N et al. haematologica 2019, 104, 564; Singh-SK et al. Sci Reports 2018, 8, 1323). Is the CCR5-MIP-1a interaction essential for the enhancement of CD4+ and CD8+ antitumor capacity? Is the action of maraviroc on T cells as well as on tumor cells essential for its antitumor potential? These questions need to be explained.

Page 10, lines 286-289: (f) is incorrectly located. (i) does not exist in Figure 2. Needs to be corrected.

Page 14, line 364: There is no subtitle for 3.5.

Duplication is observed in references.

  References 22 and 31,

  References 29, 35 and 37.

Author Response

Dear Reviewer 1,

Thank you for your careful consideration and evaluation of our manuscript, please find our rebuttal letter attached where we address your comments,

Thank you for your insights and feedback,

Best wishes,

Maria Davern

Reviewer 2 Report

Comments and Suggestions for Authors

This is a well-written manuscript summarizing the recent progress from continuous work. The authors analyzed the tumor immune microenvironment and identified that CCR5 may contribute to immunosuppression. Most of the data is well presented. The only concern is that based on the results in Figures 4 & 5, it may not be described as “the synergistic effects of radiation plus CCR5 antagonism on T cell function and migration”, it is more like CCR5 antagonism may improve radiation therapy or radiation resistance.

Author Response

Dear Reviewer 2,

Thank you for your careful consideration and evaluation of our manuscript, please find our rebuttal letter attached where we address your comments,

Thank you for your insights and feedback,

Best wishes,

Maria Davern

Round 2

Reviewer 1 Report

Comments and Suggestions for Authors

There are no additional comments.